# Investigating the relation between self-assessment and patients' assessments of physicians-in-training empathy: a multicentric, observational, cross-sectional study in three teaching hospitals in Brazil

Mônica Oliveira Bernardo,[1] Dario Cecilio-Fernandes,[2]
Alba Regina de Abreu Lima,[3] Julian Furtado Silva,[4] Hugo Dugolin Ceccato,[4]
Manuel João Costa,[5] Marco Antonio de Carvalho-Filho[2,4]

MOB and DC-F contributed equally.

For numbered affiliations see end of article.

**Correspondence to**
Dr. Marco Antonio de Carvalho-Filho;
m.a.de.carvalho.filho@umcg.nl, macarvalhofilho@gmail.com

## ABSTRACT

**Objectives** This study investigated the associations between self-assessed empathy levels by physicians in training and empathy levels as perceived by their patients after clinical encounters. The authors also examined whether patient assessments were valid and reliable tools to measure empathy in physicians in training.

**Design** A multicentric, observational, cross-sectional study.

**Setting** This study was conducted in three public teaching hospitals in Brazil.

**Participants** From the 668 patients invited to participate in this research, 566 (84.7%) agreed. Of these, 238 (42%) were male and 328 (58%) were female. From the invited 112 physicians in training, 86 (76.8%) agreed. Of the 86 physicians in training, 35 (41%) were final-year medical students and 51 (59%) were residents from clinical and surgical specialties. The gender distribution was 39 (45%) males and 47 (51%) females.

**Primary and secondary outcome measures** Physicians in training filled the Jefferson Scale of Physician Empathy (JSE) and the Interpersonal Reactivity Index. Patients answered the Jefferson Scale of Patient's Perceptions of Physician Empathy (JSPPPE) and the Consultation and Relational Empathy Scale (CARE).

**Results** This study found non-significant correlations between patient and physicians-in-training self-assessments, except for a weak correlation (0.241, p<0.01) between the JSPPPE score and the JSE compassionate care subscore. CARE and JSPPPE scales proved to be valid and reliable instruments.

**Conclusions** Physicians-in-training self-assessments of empathy differ from patient assessments. Knowledge about empathy derived from self-assessment studies probably does not capture the perspective of the patients, who are key stakeholders in patient-centred care. Future research on the development of physician empathy or on outcomes of educational interventions to foster empathy should include patient perspectives.

---

### Strengths and limitations of this study

- ► This was a multicentric study involving three public teaching hospitals.
- ► This study combined multiple perspectives of physicians-in-training empathy.
- ► The ratio of patients to physicians in training was high, thus decreasing the bias in patient assessments, resulting in reliable empathy measurements.
- ► This study did not take into account elements that may interfere with patients' experiences like the time spent in the consultation and/or waiting room.

## INTRODUCTION

Physician empathy is crucial for patient care.[1 2] Empathy enhances professional fulfilment,[1] diminishes physicians' burn-out[3] and is related to clinical competence.[4] Empathy is vital to understand, acknowledge and address patients' needs in clinical encounters and to construct a shared and feasible therapeutic plan considering patients' beliefs and context.[5 6] Empathetic behaviour of health professionals is a cornerstone to establish meaningful conversations with patients, decrease anxiety, and reveal patients' emotions and beliefs relevant to patients' experiences.[7–9]

The relevance of empathy to patient care has led to increasing calls to develop learning strategies to foster the capacity of physicians in training for empathic understanding.[4] Unfortunately, research findings with undergraduate students have raised concerns over eventual negative impacts of medical schools on student empathy (for a discussion see Ferreira-Valente *et al*[10]). Despite the

controversy over the effect of medical school on empathy, it is consensual that an important limitation of empathy research has been the frequent reliance on participant self-assessments, in general using the Jefferson Scale of Physician Empathy (JSE)[11] or the Interpersonal Reactivity Index (IRI).[12] This is the case, for example, of longitudinal studies of medical student empathy[13] or of studies on outcomes of interventions to develop empathy.[14–16]

Despite the international use of established questionnaires, it is unknown to what extent empathy self-assessment scores reflect empathic behaviours as observed by others, especially patients.[17–19] The wider literature on self-assessments suggests these may be insufficient to capture the full breadth of behaviours or attitudes.[19 20] It is therefore of paramount importance to characterise associations between empathy assessed by oneself and by others.

Recently, a multicentre study in Brazil[19] of the associations between self-reported and patient-derived empathy measures for physicians reported that those may be unrelated, suggesting that doctor empathy self-assessments were not indicative of empathy as perceived by patients. It is thus likely that the same is the case for resident or student empathy, but this remains to be confirmed empirically. Moreover, a recent study showed a lack of correlation between self-assessed empathy by primary care physicians and clinical outcomes in patients with diabetes.[21] The lack of correlation between self-assessed empathy and both patients' assessments and clinical outcomes is a powerful argument to expand the field towards including patients' perspectives.

In the current article, we sought out to investigate the correlation between medical students' and residents' self-assessed empathy levels with the empathy levels as perceived by the patients assisted by them directly in real clinical encounters. We also investigated whether patients' measures are valid and reliable tools to be used as assessment methods of the empathy levels of medical students and residents. We believe that understanding how patients perceive the empathy of medical students and residents in the context of real medical encounters can inform educational interventions to foster a more humanistic practice. Our research may help to enlighten the utilisation of empathy measurements to guide educational practices.

## METHODS
### Context
We performed this research in three teaching public hospitals in São Paulo, Brazil, in which interns and residents are independently responsible for the clinical consultations. There are two steps in any consultation. First, interns or residents interview and perform an autonomous clinical examination of the patient. Then, they meet the supervisor in another room to establish the principal diagnostic hypothesis, the differential diagnosis and the treatment plan. Finally, they come back to their patients to share the plan. Thus, patients interact directly and exclusively with the physicians in training (interns and residents), not with the supervisors. The autonomy of interns and residents in this context creates a unique opportunity to obtain patients' perspectives on interns' and residents' empathy, with no influence of supervisors.

### Participants
Promptly after each consultation, a researcher (MOB, ARdAL, JFS or HDC) invited patients to participate in the study. The inclusion criteria were patients to be older than 18 years old, literate and capable of filling the assessment instrument. In total, we invited 668 patients to participate in the study. None of the researchers had care responsibilities for any of the patients.

We invited to participate in the study physicians in training who were either year 5 or year 6 senior medical students (interns) or years 1–3 residents from diverse specialties. In total, we invited 112 physicians in training (interns+residents) to participate in the study.

### Instruments
We used four instruments to measure empathy: two based on self-assessment and two based on direct observation by actual patients.

### Self-assessment instruments
We used the physician version of the Jefferson Scale of Physician Empathy (JSE)[11] and the IRI.[12] These are the two scales most extensively used in empathy research. The JSE was developed specifically for healthcare contexts, whereas the IRI was developed for the general population. Both instruments have a mixture of positive and negative items inviting respondents to rate the extent to which they agree or disagree with each statement.

The JSE[11] consists of 20 items rated on a 7-point Likert scale. The JSE measures three subdimensions of empathy: 'Perspective Taking', 'Compassionate Care' and 'Standing on the Patient's Shoes'.[11] All the negative items were transformed into positive to calculate the scores. The overall score for the JSE is the sum of all items' scores, and the scores of the subdimensions are the sum of their respective items.

The IRI consists of 28 items rated on a 5-point Likert scale. The IRI measures four subdimensions of empathy: 'Perspective Taking', 'Empathic Concern', 'Personal Distress' and 'Fantasy'.[12] All negative items were transformed into positive to calculate the scores. The overall score for the IRI is the sum of all items scores, and the scores of the subdimensions are the sum of their respective items.

We used the physician and student Portuguese versions of the JSE[22] and IRI.[23]

### Patients' assessment instruments
To measure physicians' empathy as perceived by their patients, we used the Jefferson Scale of Patient's Perceptions of Physician Empathy (JSPPPE)[24] and the Consultation and Relational Empathy Scale (CARE).[25]

We used the JSPPPE because it shares the same concept of empathy as the JSE—both were developed by the

same research group.[24] CARE was developed aiming the concept of empathy as perceived by patients, and there is evidence of reliability, internal validity and consistency of this scale.[26]

The JSPPPE has five items rated on a 7-point Likert scale.[24 27] The overall score for the JSPPPE is the sum of all items' scores. We applied the validated Portuguese version.[19]

The CARE instrument has 10 items rated on a 6-point Likert scale.[25 26] The overall score for CARE is the sum of all items scores. We used the validated Portuguese version of CARE.[28]

Previous studies had demonstrated that the JSPPPE and CARE were unidimensional with high reliability coefficients—respectively, 0.88 and 0.97.[19] As the Portuguese version of these instruments had only been validated with a sample of medical doctors in Brazil, we have conducted new analysis to investigate whether the psychometric properties were similar for physicians in training.

### Study procedures

The sample of physicians in training is a convenient sample since the authors (MOB, ARdAL, JFS and HDC) had to seize the opportunity of inviting participants during their working hours in the referred university hospitals. The patient sample, on the contrary, comprehended all patients consulted by their respective physicians in training on the day of selection.

Patients were informed about the aim of the research and assured that participating in the study would not affect their care. Then, patients were invited to sign the consent form. Subsequently, only patients who signed the inform consent filled the questionnaires.

Physicians in training were informed about the aim of the research and that participating in the study would not affect their assessment during the clinical rotation. Then, physicians in training were asked to sign the inform consent. Only physicians in training who signed the inform consent filled the questionnaires. The physicians in training filled the questionnaires only once before we started collecting the patients' questionnaires.

We used paper questionnaires for both patients and physicians in training. All forms were anonymised and inserted into a data system by a designated person, who did not have access to patients' names.

### Data analyses

We compared patients' assessments considering their gender, physicians' gender and physicians' training level (intern vs resident) using t-tests. We also compared physicians-in-training self-assessment of empathy in respect of their training level (intern vs resident) using t-tests.

We used Pearson correlation to investigate the relation between physicians-in-training self-assessed empathy with the empathy perceived by their patients. As the number of patients per physician in training differed (ranging from 3 to 15), we averaged patients' responses to each

physician in training before conducting the Pearson correlation analysis.

To investigate whether CARE and JSSSPE were valid and reliable tools in specific study population, we conducted a confirmatory factor analysis with maximum likelihood estimation. We calculated the reliability using Cronbach's alpha. We also investigated the concurrent validity of the scales by comparing the scores of the JSSPE and CARE using Pearson correlation.

### Patient and public involvement

Our research sought out to explore the potential contribution of patients' feedback on physicians-in-training empathy aiming a better quality of patient care and experience. We involved patients who voluntarily accepted to participate in a random selection. Patients were not involved in study design. The results of our study will be available for all the institutions and their patient representatives.

## RESULTS

From the 668 patients invited to participate in this research, 60 declined and 40 were excluded due to difficulties in completing the instruments. In total, 566 (84.7%) patients participated in this research, from three different university hospitals: university hospital A (n=237), university hospital B (n=151) and university hospital C (n=178). Of these, 238 (42%) were male and 328 (58%) were female. Patients' age ranged from 18 to 77, with a mean of 47 years old.

From the invited 112 physicians in training, 20 refused to participate and 6 were excluded because of uncorrected filling of the JSE or the IRI. In total, 86 (76.8%) physicians in training from three university medical hospitals in Brazil (university hospital A, n=36; university hospital B, n=17; university hospital C, n=33) participated in this study. Of the 86 physicians in training, 35 (41%) were interns and 51 (59%) were residents from clinical and surgical specialties. The gender distribution was 39 (45%) males and 47 (51%) females, with ages ranging from 22 to 33 years old. The residents' specialties were surgery (n=21), internal medicine (n=21) and gynaecology (n=9).

### Physicians-in-training self-assessments

Interns scored higher than residents in both the JSE and IRI scales, but the differences were only statistically significant for the JSE. Female physicians scored significantly higher in both the JSE and the IRI (table 1).

We found a moderate and significant correlation between the JSE and the IRI (r=0.44, p<0.05). We also found positive and significant correlations between the subscales ranging from weak to moderate magnitude (table 2).

### Patient assessments

Interns scored significantly higher than residents on both empathy patients' scales. Female physicians in training

**Table 1** Descriptive and comparative statistics for empathy self-assessment by physicians in training

| | | n | JSE (SD) | P value | IRI (SD) | P value |
|---|---|---|---|---|---|---|
| Physicians in training | Intern | 35 | 121.14 (9.52) | <0.05 | 67.17 (11.56) | >0.05 |
| | Resident | 51 | 114.22 (14.26) | | 65.18 (14.02) | |
| Physicians' gender | Male | 39 | 112.90 (14.6) | <0.01 | 59.56 (13.33) | <0.001 |
| | Female | 47 | 120.47 (10.35) | | 71.32 (10.18) | |

IRI, Interpersonal Reactivity Index; JSE, Jefferson Scale of Physician Empathy.

scored significantly higher on the JSPPPE scale but not on CARE. There were no differences in empathy scores according to patients' gender (table 3).

### Associations between patients' assessments and self-assessments of empathy

We did not find any correlation between the total scores of patients' and self-assessment scales. The same was true for the subdimensions of the scales, with one only exception. There was a positive and weak correlation of the JSPPPE score with the JSE compassionate care subscore (table 4).

### Validity and reliability of JSPPPE and CARE

The base model of confirmatory factor analysis for the JSPPPE scale (model A) displayed a moderate fit index values, based on the Tucker-Lewis index (TLI), comparative fit index (CFI) and root mean square error of approximation (RMSEA). In model B, after we added the correlation between the items' errors, the model reached a satisfactory level of model fit (table 5), demonstrating evidence of validity for the JSPPPE. Cronbach's alpha was 0.91, indicating that the instrument is reliable.

The base model of confirmatory factor analysis for the CARE (model A) displayed a moderate fit index values, based on TLI, CFI and RMSEA. In model B, after we added the correlation between the items' errors, the model reached a satisfactory level of model fit (table 5), demonstrating evidence of validity for CARE. Cronbach's alpha was 0.96, indicating that the instrument is reliable.

### DISCUSSION

In this study, we sought to investigate whether empathy self-assessment by physicians in training correlated with their patients' assessments. We also evaluated the validity and reliability of the two instruments for patients' assessments of physicians in training. Our findings corroborated the hypothesis that self-assessment of empathy by interns and residents did not correlate with patients' assessments, in line with findings with senior clinical practitioners.[19] Taking into consideration that this study used four empathy scales—two self-assessments and two patient assessments—most widely used on empathy research, the findings are of particular relevance. A former study in five countries had demonstrated that the two self-reported scales did not capture the same empathy construct.[29] The lack of correlation between self-assessed and patient-assessed empathy has implications for how the literature on health professionals' empathy is interpreted. Quite likely, findings from studies using any of the four scales, which often compare empathy across studies, are not directly comparable. Like in the parabola of the elephant and the blind man in which each blind man feels a different piece of the elephant, it is possible that such studies capture different elements of the complex psychological construct called empathy. To develop empathy studies relevant to inform medical education, it is crucial, at this moment, to clarify which scale—if any—offers measure which correlates with meaningful clinical or educational outcomes. For example, recently, Chaitoff et al[21] found that self-assessed empathy levels of primary care physicians were not correlated with laboratorial outcomes in patients with diabetes. This result enlightens the debate on the correlation between self-assessed empathy and clinical outcomes by showing that a relationship of cause–effect between those two variables is unlikely.

**Table 2** Pearson correlations between JSE and IRI

| | | IRI | | | | |
|---|---|---|---|---|---|---|
| | | Fantasy scale | Perspective taking | Empathic concern | Personal distress | IRI total |
| JSE | Perspective taking | 0.355* | 0.285* | 0.632* | −0.048 | 0.485* |
| | Compassionate care | 0.364* | 0.342* | 0.603* | 0.046 | 0.346* |
| | Standing in the patient's shoes | 0.318* | 0.184 | 0.492* | −0.035 | 0.031 |
| | Jefferson total | 0.033 | 0.038 | 0.240* | −0.183 | 0.435* |

*p<0.05

IRI, Interpersonal Reactivity Index; JSE, Jefferson Scale of Physician Empathy.

**Table 3** Descriptive and comparative statistics for empathy measurements by patients

| | | n | JSPPPE (SD) | P value | CARE (SD) | P value |
|---|---|---|---|---|---|---|
| Physicians in training | Intern | 191 | 33.27 (3.59) | <0.001 | 46.37 (6.95) | <0.001 |
| | Resident | 375 | 29.81 (6.68) | | 41.81 (8.09) | |
| Physicians' gender | Male | 276 | 30.04 (6.72) | <0.001 | 43.04 (8.25) | >0.05 |
| | Female | 290 | 31.87 (5.18) | | 43.63 (7.8) | |
| Patients' gender | Male | 238 | 31.31 (5.78) | >0.05 | 43.23 (8.3) | >0.05 |
| | Female | 328 | 30.73 (6.22) | | 43.43 (7.8) | |

CARE, Consultation and Relational Empathy Scale; JSPPPE, Jefferson Scale of Patient's Perceptions of Physician Empathy.

Empathy is a complex construct with cognitive, affective, behavioural and moral dimensions entailing different lenses to be fully understood.[1 30] When patients are invited to discuss what is a positive outcome through their perspectives, the complexity increases. Patients' definition of a positive outcome may vary along the course of their disease and life. We believe that empathy is a necessary psychological trait for the doctor to understand the singularity of each patient and individualise therapeutic plans in alignment with patients' needs and beliefs. Considering both the complexity of empathy and the singularity of patients' experiences, we invite researchers in this field to expand their focus.

First, longitudinal studies or pretest/post-test evaluations of learning strategies using self-reported empathy cannot anticipate physicians-in-training performance on real clinical encounters. We agree that self-reported measurements can help teachers to start the conversation with students around the relevance of being empathic for becoming a caregiver. However, if educators want to mirror future performance, it is crucial to include real

**Table 4** Pearson correlations between empathy measurements: self-assessments versus patients' assessments

| | | Patients' perceptions (n=566) | |
|---|---|---|---|
| Physicians' perceptions (n=86) | | JSPPPE | CARE |
| JSE | Perspective taking | 0.011 | 0.168 |
| | Compassionate care | 0.241* | 0.207 |
| | Standing in the patient's shoes | 0.109 | 0.033 |
| | Jefferson total | 0.149 | 0.196 |
| IRI | Fantasy scale | −0.013 | 0.172 |
| | Perspective taking | 0.066 | −0.067 |
| | Empathic concern | 0.083 | 0.044 |
| | Personal distress | 0.011 | 0.047 |
| | IRI total | 0.046 | 0.089 |

*p<0.05
CARE, Consultation and Relational Empathy Scale; IRI, Interpersonal Reactivity Index; JSE, Jefferson Scale of Physician Empathy; JSPPPE, Jefferson Scale of Patient's Perceptions of Physician Empathy.

patients in the assessment of students. Specially during the transition to independent practice, when students face the challenges related to adapting to the constraints of the healthcare system, medical educators should reinforce the importance of empathy, while helping students to align theory and practice through effective role-modelling.[31]

Second, studies targeting the understanding of the possible association between empathy and clinical outcomes should take into consideration the importance of empathy on building a therapeutic alliance between the doctor and the patient. So we hypothesise that punctual assessments by patients after a singular encounter are not enough to capture the phenomena under study and we should use instruments to measure the quality of the relationship between the doctor and patient. However, we do believe that punctual assessments may be efficient to give feedback to clinicians on their attitudes and behaviours, nurturing the development of their communication skills.

As expected, our results demonstrated that self-assessment of empathy by interns and residents does not correlate with patients' assessments, in alignment with the results previously observed for senior clinical practitioners.[19] This finding corroborates the general literature that points out the inaccuracy of self-assessment.[17 18] Overall, physicians in training might become overconfident over time. For example, consecutive participation in clinical practice may increase students' self-confidence,[32 33] which does not necessarily predict their performance. Furthermore, self-assessment questionnaires in empathy often focus only on whether the participants are aware of how they have to behave to be empathic. However, knowing how to behave does not necessarily translates into a change of the behaviour in practice. Our results are aligned with this possible mismatch between intention and action.

Our results also suggest that the CARE and JSPPPE scales could be used as assessment tools and detect elements that may interfere in patients' perception of students and residents' empathy. More importantly, the psychometric properties of the instrument are very similar when looking at the physicians in training and medical doctors.[19] Both CARE and JSPPPE have followed the same internal structure and similar reliability coefficient. Contradictory to previous studies where weak correlations were found between the JSE

**Table 5** Fit index for the JSPPPE and CARE

| | | $\chi^2$ (df), significance | TLI | CFI | RMSEA (HI90) |
|---|---|---|---|---|---|
| JSPPPE | Model A | $\chi^2$ (5)=30.177, p<0.001 | 0.975 | 0.987 | 0.094 (0.128) |
| | Model B | $\chi^2$ (4)=15.501, p=0.004 | 0.986 | 0.994 | 0.071 (0.110) |
| CARE | Model A | $\chi^2$ (35)=204.716, p<0.001 | 0.960 | 0.969 | 0.093 (0.105) |
| | Model B | $\chi^2$ (26)=51.538, p=0.002 | 0.992 | 0.995 | 0.042 (0.058) |

CARE, Consultation and Relational Empathy Scale; CFI, Comparative Fit Index; HI90, upper limit 90% confidence interval; JSPPPE, Jefferson Scale of Patient's Perceptions of Physician Empathy; RMSEA, root mean square error of approximation; TLI, Tucker-Lewis index.

and the IRI,[29] our results indicated a moderated correlation between the JSE and the IRI, suggesting that both scales may be measuring the same aspect of empathy. Although this finding adds to the number of validity evidence of the JSE, the outcomes should be carefully interpreted since it seems that the relation may be related to the sample and context.

Patients found interns more empathic than residents, which raises a concern about the effects of the transition to practice on physicians-in-training empathy levels. Previous studies have shown a decline in self-assessed empathy levels during undergraduate medical training,[34] but this finding is not universal across different medical schools.[10] Studies evaluating self-assessed empathy levels during residency training showed heterogeneous results.[35–37] Our study was not designed to investigate the evolution of empathy throughout the maturation of doctors, but it raises the possibility that empathy as perceived by patients can decrease during the transition from internship to residency training. Patients also considered female interns and residents to be more empathetic than males, a phenomenon that is also observed with self-assessed empathy.[38]

Although our study had a cross-sectional design, it corroborates the importance of including patients' perspectives into this debate. Without giving voice to patients, we will not have a comprehensive understanding of how medical training affects empathy development of students and residents. Without patients' insights, we will also struggle to realise whether our pedagogical interventions are impacting students the way we have planned.

A limitation of the present study is that we did not address the influence on patient perceptions of contextual or environmental factors, such as the consultation time, delay in the waiting room or the comfort of the environment. These elements may have influenced patients' perspectives.[39 40] Another limitation was the study's inability to pair in time the assessments of physicians and patients. Also, the sample of physicians in training was not randomly selected. A final limitation was that participants were informed of the nature of the research, which may have induced behaviours more socially desirable and have biased the results.

The relatively large number of patients is one of the strengths of this study. Such a high number of patients allowed us (1) to decrease the bias, which may occur when one patient may have a different perspective of the

others, and (2) to obtain a reliable measurement of the level of empathy of the physician in training.

This study adds evidence to the complexity of measuring empathy. The observed mismatch raises the question of whether educational interventions to foster empathy should rely solely or preferably on self-assessment measurements to attest their quality or relevance. Although self-assessment may function as a stimulus to create awareness and motivation to change in trainees, patients' perspectives are crucial to improve the actual care and to verify the efficiency of pedagogical interventions.[19]

Patients' assessments are a meaningful opportunity to engage trainees in a reflection on the relevance of developing themselves into empathic caregivers.[41 42] Inviting patients to share their perspectives allows physicians in training to gather feedback from the people they intend to care for, the people they should strive to understand, reassure and advise.[43] Furthermore, empowering patients as formal assessors reinforces the message that a good doctor acknowledges, reflects on and reacts to patients' opinions and views.

If our ultimate goal is to increase physicians' empathy towards the patient, the assessment methods applied to evaluate the empathy levels of medical students and residents should include patients' perspectives. Including patients' perspectives paves the way for educational interventions to impact the reality of practice, which is the ultimate goal of medical education.

Concluding, our study demonstrated a mismatch between physicians-in-training empathy self-assessment and their patients' assessments. This finding may have two implications: (1) patients' instruments may be measuring a different component of empathy, and (2) the self-assessment of empathy probably is not enough to foster more humanistic patient care.

**Author affiliations**
[1]Radiology, Pontificia Universidade Catolica de Sao Paulo Faculdade de Ciencias Medicas e da Saude, Sorocaba, São Paulo, Brazil
[2]CEDAR - Center for Education Development and Research in Health Professions, University Medical Center Groningen, Groningen, The Netherlands
[3]Medical Faculty of São Jose do Rio Preto, São José do Rio Preto, Brazil
[4]Internal Medicine, Universidade Estadual de Campinas Faculdade de Ciencias Medicas, Campinas, São Paulo, Brazil
[5]Life and Health Sciences Research Institute, School of Health Sciences, University of Minho, Braga, Portugal

**Acknowledgements** We thank all the patients, patients' advisors, interns and residents who allowed us to collect the data that support our study.

**Contributors** MOB, MJC and MAdC-F have substantially contributed to the design of the work. MOB, ARdAL, JFS and HDC were responsible for the acquisition of data. DC-F was responsible for the data analyses. MOB, DC-F, MJC and MAdC-F were responsible for the interpretation of the data. MOB and DC-F were responsible for the first draft of the paper. All the authors have critically revised and approved the final version of the paper. All authors are accountable for all aspects of the work.

**Funding** This study was funded by the 'Fundação de Amparo à Pesquisa do Estado de São Paulo - FAPESP' (grant number: 2016/11908-1) and by the 'Conselho Nacional de Desenvolvimento Científico e Tecnológico - CNPq' (grant number: 202319/2017-2).

**Competing interests** None declared.

**Patient consent for publication** Not required.

**Ethics approval** We obtained ethical approval for this study from the Research Ethics Committee of the three universities involved (university A, CAAE=63847016.90.1001.5373; university B, CAAE=63847016.90.2002.5404; and university C, CAAE=63847016.90.2001.5415). All participants gave written informed consent before data collection.

**Provenance and peer review** Not commissioned; externally peer reviewed.

**Data sharing statement** All data relevant to the study are included in the article or uploaded as supplementary information.

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
