## [Reviewer comments · BMJ Open]

ARTICLE DETAILS

TITLE (PROVISIONAL)	Investigating the relation between self-assessment and patients' assessments of physicians in training empathy: a multicentric observational cross-sectional study in three teaching hospitals in Brazil
AUTHORS	Bernardo, Mônica; Cecilio-Fernandes, Dario; Lima, Alba; Silva, Julian; Ceccato, Hugo; Costa, Manuel; de Carvalho-Filho, Marco Antonio

VERSION 1 - REVIEW

REVIEWER	Kristen Kim Rutgers New Jersey Medical School, USA
REVIEW RETURNED	26-Feb-2019

GENERAL COMMENTS	Thank you for this interesting and important study on the relationship between self and patient assessment of empathy among physicians in training. I agree with the conclusion that the findings support the incorporation of patient assessments in medical education. My main recommendation is that the authors improve the overall grammar and punctuation of the manuscript. For example, the title could be simplified and sentence structure should be corrected to "between _ and _," e.g., "between self-assessment of empathy by physicians in training and patients' assessments..." Such errors can be seen throughout the manuscript and significantly hinder readability. The abstract overall provides a good overview of the study. One suggestion is to either exclude mention of the discrepancy between medical student and resident empathy scores or to include a line about a possible interpretation in the conclusion. Another suggestion is to re-order the strengths and limitations such that strengths are presented first and then limitations. I recommend keeping the order of assessments consistent throughout the sections. In the methods section, the instruments are listed as self-assessment ones first and then patient's assessment ones. In the results section, however, the order is switched and the patient assessments are presented first followed by the self-assessments.
--

	In the discussion section, the authors can consider including further interpretation of the gender differences and intern versus resident differences mentioned in the results section. In other words, what do the authors make of the higher empathy scores among females? How do these results relate to existing research? Additionally, what is the existing research on the change in empathy throughout medical training and how do the authors' findings compare? In regard to publication ethics, under Conflicts of Interests, the manuscript reads, "The authors declare any conflict of interests." Please clarify.
--	---

REVIEWER	Léonore Robieux Institut Curie, France
REVIEW RETURNED	01-Mar-2019

GENERAL COMMENTS	I was thrilled to see research on students empathy. Introduction:  1. The authors cite one of their paper (Bernardo et al., 2018) adopting a design equivalent to that of this paper but with a population of doctors and not of trainees. So I wonder if the paper being revised could be improved from a comparison between these two populations and the more indicative results. 2. p7. l13-15: This formulation gave me the impression that we were going to explore other factors besides empathy such as organizational factors as suggested in discussion. 3. If the objective is to identify the heteroevaluation measures as valid for the student population, then it seems all the more important to me to make a comparison with a population to which this tool is validated and recognized. Methods:  1. Some information on the psychometric qualities of the tools used is expected, in particular to justify the choice of these tools. (Can the factorial validity of the JSE be at the origin of the results?) 2. the physician sampling method represents a definite bias that could be limited by adding participants through a snowball technique. 3. What led the authors to choose not to use the JSPE student version? 4. When and how often do the trainees complete the questionnaires? In longitudinal studies, physicians are often asked to fill the empathy scale at each time because different authors have identified its situational dimension. Results  1. May the authors indicate the reasons of declined of patients and trainees? 2. The evaluation of internal fidelity and the factorial validity of the empathy heteroevaluation scales is too unjustified and is surprising since the tool validation paper, other studies already using these measures have reported it? Is this choice motivated by the specific physician population? The contribution of this study is really the in vivo evaluation crossed with the students.
---

	3. An evaluation of the concurrent validity between the tools of the patients or between the JSE and the JSPPE could be interesting? Discussion :  1. There is little discussion of psychometric results. 2. One possible bias in the results is that the assessments of physicians and patients are not given in pairs, ie an assessment of the level of empathy following each encounter with a patient compared to the score given by the patient. Authors consider different factors as the time of consultation as possible element to model to explain these differences. However, it is possible to consider the level of empathy of a trainee as identical from one patient to another, from one moment to another of the day, from one medical situation to another is somewhat reducer
--	---

REVIEWER	Alexander Chaitoff Cleveland Clinic Lerner College of Medicine of Case Western Reserve University, Cleveland, Ohio, United States
REVIEW RETURNED	04-Mar-2019

GENERAL COMMENTS	This study utilized a cross-sectional design to explore the relationship between medical student/resident self-reported vs patient-reported measures of empathy. They found that the self-report measures did not tend to correlate with the patient-reported measures, and that the self-report measures did have construct validity. The study is incredibly important, as currently there are many ways that “empathy” is studied in the literature, but little evidence about how those measures relate to each other or which ones have clinical relevance. The study has several notable strengths, including the important question it addresses, the relatively large sample size of patients, and use of multiple scales. However, the study does have several areas for improvement, most notably that the discussion is relatively short and the discoveries the authors’ make are not contextualized. Thank you very much for the opportunity to review this manuscript. Section-specific comments are below: Introduction -Page 6 Line 5. The background sections of papers often simply state that physician empathy correlates with outcomes, but studies are increasingly demonstrating that physician characteristics (Tuerk et al., 2008, Diabetes Care), and more specifically physician empathy (Chaitoff et al., 2019, Journal of General Internal Medicine), may not be nearly as strongly associated with patient satisfaction or outcomes as once believed (at least not when empathy is self-reported, as is the case in many studies underpinning the empathy literature). Articles studying empathy need to avoid hyperbole in describing the importance of “physician empathy” (defined in any number of ways) in order to avoid becoming perspective pieces. I would encourage the authors to not simply cite entire books (written by members of a research group that has a clear financial conflict of interest with the JSE) or
--

reviews to support their argument that empathy is important. Instead, they should make their argument in their introduction by including more quality primary literature while also noting the very real limits of our understanding of how empathy as it is currently measured impacts patient experience and clinical outcomes. Given the very real limitations of self-report empathy (namely that it is being increasingly shown not to correlate with meaningful outcomes), this is necessary to explain why this study of patient-report empathy is so important.

-Page 6 Lines 31-54 (paragraph 2). This paragraph is difficult to follow and seems to try to explain to the reader why therapeutic plans succeed or fail, which is beyond the scope of this paper. I would shorten this considerably or remove it entirely.

-Page 7 Lines 29-36. Much of these sentences appears to be more appropriate for the discussion, I would remove from the introduction.

Methods

-I think a brief discussion of why the 4 scales were chosen would be helpful. Specifically, of the self-report empathy measurements, IRI was not developed to necessarily be healthcare specific as compared with the JSE, which was developed for use in healthcare populations. However, both the JSPPE and CARE patient-report empathy measures were developed with healthcare settings in mind. Why not select one of each type (a healthcare specific scale and one more general scale of empathy) for both self-report and other-report measures of empathy?

-The authors should add an analysis. Specifically, they should calculate correlation coefficients for the IRI vs JSE. To my knowledge, in Hojat et al.'s validation papers on the JSE, they never compare JSE to an accepted standard (like the IRI), and studies by other authors (eg Costa et al., 2017, Academic Medicine) that have done so have reported weak correlations between the two. If it is found that so much of the physician empathy literature may be based on a scale that doesn't measure empathy as it is most commonly conceived by others in psychology/sociology, it would be very pertinent to report here.

-I wonder if the pearson correlation should be spearman correlations. From my experience, empathy scores are often not normally distributed; the authors might want to check a histogram of empathy scores to ensure appropriateness of the analysis.

-Page 10 Line 10-11. Physicians were informed of the nature of the research, which may bias the results (social desirability bias). This should be mentioned as a limitation in the discussion.

Results

-Would add two columns to table three to compare the JSE and IRI to each other (as described above).

-Would appreciate measures of spread be included throughout the results (SD or IQR, whichever is more appropriate).

Discussion

-I think the discussion about the inaccuracy of self-assessment should be expanded, perhaps discussing why the mismatch occurs.

-Page 15, line 34. The authors state that the "CARE and JSPPE scales could be used as assessment tools and detect elements that may interfere in patients' perception of students and residents empathy." I would like more of a discussion of why this is an appropriate recommendation. This study shows that the JSPPE

	and CARE do not correlate with the IRI (a well-validated scale of empathy), and, to my knowledge, there has not been convincing evidence showing that either scale correlates with any meaningful clinical outcomes. As such, while the scales may have construct validity, I'm not sure what the purpose of using them in medical education assessment would be. If, however, the scales have been shown to correlate with a meaningful outcome (be they patient experience or clinical in nature), then that should be reported here in the discussion. This discussion might also touch on the bigger issue with the empathy literature – lots of correlations with communication/patient experience outcomes more generally, but not many scales shown to correlate with clinical outcomes. -The correlation between JSE and IRI should be reported in the results and a discussion of what is found should be reported in the discussion. -The limitations can be expanded, including mentioning potential effects of discussing the purpose of the study with participants prior to providing them the scales.
--	---

VERSION 1 – AUTHOR RESPONSE

Reviewer(s)' Comments to Author:

Reviewer: 1

Reviewer Name: Kristen Kim

Thank you for this interesting and important study on the relationship between self and patient assessment of empathy among physicians in training. I agree with the conclusion that the findings support the incorporation of patient assessments in medical education.

Commentary 1: My main recommendation is that the authors improve the overall grammar and punctuation of the manuscript. For example, the title could be simplified and sentence structure should be corrected to “between _ and _,” e.g., "between self-assessment of empathy by physicians in training and patients' assessments..." Such errors can be seen throughout the manuscript and significantly hinder readability.

Thank you very much for your comment. We have thoroughly revised the overall grammar and punctuation.

Our new title is “Investigating the relation between self-assessment and patients` assessments of physicians in training empathy: a multicentric observational cross-sectional study in three teaching hospitals in Brazil.”

Commentary 2: The abstract overall provides a good overview of the study. One suggestion is to either exclude mention of the discrepancy between medical student and resident empathy scores or to include a line about a possible interpretation in the conclusion. Another suggestion is to re-order the strengths and limitations such that strengths are presented first and then limitations.

We agree with the reviewer that we should exclude from the abstract the observed discrepancy between students and residents. We also have re-ordered the strengths and limitations and reviewed the abstract.

The new abstract and strengths and limitations session are as follows:

“ABSTRACT

Objectives: This study investigated the associations between self-assessed empathy levels by physicians in training and empathy levels as perceived by their patients after clinical encounters. The authors also examined whether patient assessments were valid and reliable tools to measure empathy in physicians in training.

Design: A multicentric observational cross-sectional study.

Setting: This study was conducted in three public teaching hospitals in Brazil.

Participants: From the 668 patients invited to participate in this research, 566 (84,7%) agreed. Of those, 238 (42%) were male, and 328 (58%) female patients. From the invited 112 physicians in training, 86 (76.8%) agreed. Of the 86 physicians in training, 35 (41%) were final years medical students and 51 (59%) were residents from clinical and surgical specialties. The gender distribution was 39 (45%) males and 47 (51%) females.

Primary and secondary outcome measures: Physicians in training filled the Jefferson Scale of Physician Empathy (JSE) and the Interpersonal Reactivity Index (IRI). The patients answered the Jefferson Scale of Patient’s Perceptions of Physician Empathy (JSPPE) and the Consultation and Relational Empathy Scale (CARE).

Results: This study found non-significant correlations between patient and physicians in training self-assessments, except for a weak correlation (0,241, $p < 0.01$) between the JSPPE score and the JSE Compassionate Care sub-score. CARE and JSPPE scales proved to be valid and reliable instruments.

Conclusions: Physicians in training self-assessments of empathy differ from patient assessments. Knowledge about empathy derived from self-assessment studies probably does not capture the perspective of the patients, who are key stakeholders in patient centered care. Future research on the development of physician empathy or on outcomes of educational interventions to foster empathy should include patient perspectives.

Strengths and limitations of this study:

- This was a multicentric study involving three public teaching hospitals.
- This study combined multiple perspectives of physicians in training empathy.
- The ratio of patients to physicians in training was high, thus decreasing the bias in patient assessments, resulting in reliable empathy measurements.
- This study did not take into account elements that may interfere with patients’ experiences like the time spent in the consultation and/or waiting room.”

Commentary 3: I recommend keeping the order of assessments consistent throughout the sections. In the methods section, the instruments are listed as self-assessment ones first and then patient’s assessment ones. In the results section, however, the order is switched and the patient assessments are presented first followed by the self-assessments.

We have re-structured the results section to keep the order consistent throughout the sections.

In the discussion section, the authors can consider including further interpretation of the gender differences and intern versus resident differences mentioned in the results section. In other words, what do the authors make of the higher empathy scores among females? How do these results relate

to existing research? Additionally, what is the existing research on the change in empathy throughout medical training and how do the authors' findings compare?

We thank the reviewer for giving us the opportunity to explore further the discussion on the evolution of empathy throughout medical training. We have added the following paragraph to the discussion session:

“Patients found interns more empathic than residents, which raises a concern about the effects of the transition to practice on physicians in training empathy levels. Previous studies have shown a decline in self-assessed empathy levels during undergraduate medical training³⁴, but this finding is not universal across different medical schools¹⁰. Studies evaluating self-assessed empathy levels during residency training showed heterogeneous results.³⁵⁻³⁷ Our study was not designed to investigate the evolution of empathy throughout the maturation of doctors, but it raises the possibility that empathy as perceived by patients can decrease during the transition from internship to residency training. Patients also considered female interns and residents to be more empathetic than males, a phenomenon that is also observed with self-assessed empathy.³⁸”

In regard to publication ethics, under Conflicts of Interests, the manuscript reads, “The authors declare any conflict of interests.” Please clarify.

We agree with the reviewer that the conflicts of interests statement needs clarification. We have revised the sentence.

The new sentence reads as follow: “The authors declare no conflict of interests.”

Reviewer: 2

Reviewer Name: Léonore Robieux

I was thrilled to see research on students empathy.

Introduction:

1. The authors cite one of their paper (Bernardo et al., 2018) adopting a design equivalent to that of this paper but with a population of doctors and not of trainees. So I wonder if the paper being revised could be improved from a comparison between these two populations and the more indicative results.

We thank the reviewer for the commentary. We decided to make a more explicit comparison between the two studies in the introduction and also on the discussion of this paper adding the following sentences to the introduction and discussion sessions of the paper:

Introduction: “Recently, a multi-centre study in Brazil¹⁹ of associations between self-reported and patient derived empathy measures for physicians, reported that those may be unrelated, suggesting that doctor empathy self-assessments were not indicative of empathy, as perceived by patients. It is thus likely that the same is the case for resident or student empathy, but this remains to be confirmed empirically. Moreover, a recent study showed a lack of correlation between self-assessed empathy by primary care physicians and clinical outcomes in patients with Diabetes²¹. The lack of correlation between self-assessed empathy and both patients’ assessments and clinical outcomes are powerful arguments to expand the field towards including patients’ perspectives.”

Discussion: “Our findings corroborated the hypothesis that self-assessment of empathy by interns and residents did not correlate with patients’ assessments, in line with findings with senior clinical practitioners.¹⁹”

2. p7. 113-15: This formulation gave me the impression that we were going to explore other factors besides empathy such as organizational factors as suggested in discussion.

We have changed the introduction as suggested by reviewer 3 and this segment was excluded from the revised text.

3. If the objective is to identify the heteroevaluation measures as valid for the student population, then it seems all the more important to me to make a comparison with a population to which this tool is validated and recognized.

Thank you for your comment. We agree with the reviewer's comment that the instruments should follow the same psychometric properties regardless of the population. Indeed, our results showed similar internal structure and reliability coefficients as the primary population. We added this comparison in the Discussion section. Conversely, the validation study was only conducted to verify whether both scales could be used for students as well.

Methods:

1. Some information on the psychometric qualities of the tools used is expected, in particular to justify the choice of these tools. (Can the factorial validity of the JSE be at the origin of the results?)

We have added information regarding the internal structure and reliability coefficient in the Method section.

"We used JSPPE because it shares the same concept of empathy as the JSE – both were developed by the same research group.²⁴ CARE was developed aiming the concept of empathy as perceived by patients and there is evidence of the reliability, internal validity and consistency of this scale.²⁶"

"Previous studies had demonstrated that JSPPE and CARE were unidimensional with high reliability coefficients – respectively 0.88 and 0.97¹⁹ As the Portuguese version of these instruments had only been validated with a sample of medical doctors in Brazil, we have conducted new analysis to investigate whether the psychometric properties were similar for physicians in training."

"We used the Physician and Student versions of the Jefferson Scale of Empathy (JSE),¹¹ and the Interpersonal Reactivity Index (IRI).¹² These are the two scales most extensively used in empathy research. JSE was developed specifically for healthcare contexts whereas IRI was developed for the general population."

2. the physician sampling method represents a definite bias that could be limited by adding participants through a snowball technique.

Thank you very much for your comment. Although convenience samples may represent a sampling bias, in vivo studies are difficult to conduct and use different sample methods. In addition, we believe that because we used three different teaching hospitals, the sampling bias would decrease because of the difference in context. Also, the sampling bias may have decreased because of our sample size. Nevertheless, we acknowledged this limitation in the discussion session.

3. What led the authors to choose not to use the JSPE student version?

As in our context the clinical consultation in the hospitals is very similar between physicians in training and medical doctors, we have decided to use the JSE and not the students' version. The main difference is that the physicians in training has to report back to their supervisor to check the diagnoses and treatment plan.

4. When and how often do the trainees complete the questionnaires? In longitudinal studies, physicians are often asked to fill the empathy scale at each time because different authors have identified its situational dimension.

In our study, physicians in training only filled the scale once. We have clarified this topic on the Method session and added it to the limitations of the study.

Method: "The physicians in training filled the questionnaires only once before we started collecting the patients' questionnaires."

Discussion: "Another limitation was the study's inability to pair in time the assessments of physicians and patients."

Results

1. May the authors indicate the reasons of declined of patients and trainees?

Since participation was voluntary, we did not ask their reasons to not participate in the study. Also, we were careful to spare all invited participants (including patients and trainees) from all kinds of pressure that could harm their will to take part on this study.

2. The evaluation of internal fidelity and the factorial validity of the empathy heteroevaluation scales is too unjustified and is surprising since the tool validation paper, other studies already using these measures have reported it? Is this choice motivated by the specific physician population? The contribution of this study is really the in vivo evaluation crossed with the students.

Response: We agree that the main contribution of this paper is the in vivo evaluation of empathy levels by the physicians in training and their patients. However, we considered important to ensure that the scales were also appropriated for physicians in training, as the validation paper had focused on a physician population.

We made this choice clear in the Method session: "Previous studies had demonstrated that JSPPE and CARE were unidimensional with high reliability coefficients – respectively 0.88 and 0.97¹⁹ As the Portuguese version of these instruments had only been validated with a sample of medical doctors in Brazil, we have conducted new analysis to investigate whether the psychometric properties were similar for physicians in training."

3. An evaluation of the concurrent validity between the tools of the patients or between the JSE and the JSPPE could be interesting?

The correlation between the JSE and JSPPE is presented in Table 4. We also have conducted a correlation between CARE and JSPPE. However, the outcomes demonstrated weak and not significant correlation between both scales ($r=0.073$, $p > 0.05$), indicating that there is no concurrent validity between CARE and JSPPE.

We added: We did not find any correlation between the total scores of patients' and self-assessment scales. The same was true for the subdimensions of the scales, with one only exception. There was a positive and weak correlation of the JSPPE score with the JSE Compassionate Care sub-score (Table 4).

Discussion:

1. There is little discussion of psychometric results.

Thank you very much for your suggestion. We have added one paragraph discussing the psychometric results of our studies.

We added the following paragraph to the discussion: "More importantly, the psychometric properties of the instrument are very similar when looking at the physicians in training and medical doctors.¹⁹ Both CARE and JSPPPE have followed the same internal structure and similar reliability coefficient. Contradictory to previous studies where weak correlations were found between JSE and IRI,²⁹ our results indicated a moderated correlation between the JSE and IRI, suggesting that both scales may be measuring the same aspect of empathy. Although this finding adds to the number of validity evidence of the JSE, the outcomes should be carefully interpreted since it seems that the relation may be related to the sample and context."

2. One possible bias in the results is that the assessments of physicians and patients are not given in pairs, ie an assessment of the level of empathy following each encounter with a patient compared to the score given by the patient. Authors consider different factors as the time of consultation as possible element to model to explain these differences. However, it is possible to consider the level of empathy of a trainee as identical from one patient to another, from one moment to another of the day, from one medical situation to another is somewhat reducer

We added this as a limitation of our study.

Discussion: "Another limitation was the study's inability to pair in time the assessments of physicians and patients."

Reviewer: 3

Reviewer Name: Alexander Chaitoff

Institution and Country: Cleveland Clinic Lerner College of Medicine of Case Western Reserve University, Cleveland, Ohio, United States

Please state any competing interests or state 'None declared': None declared

Please leave your comments for the authors below

This study utilized a cross-sectional design to explore the relationship between medical student/resident self-reported vs patient-reported measures of empathy. They found that the self-report measures did not tend to correlate with the patient-reported measures, and that the self-report measures did have construct validity.

The study is incredibly important, as currently there are many ways that "empathy" is studied in the literature, but little evidence about how those measures relate to each other or which ones have clinical relevance.

The study has several notable strengths, including the important question it addresses, the relatively large sample size of patients, and use of multiple scales.

Thank you very much for the compliments on our work.

However, the study does have several areas for improvement, most notably that the discussion is relatively short and the discoveries the authors' make are not contextualized.

Thank you very much for the opportunity to review this manuscript. Section-specific comments are below:

Thank you for your thorough evaluation of the manuscripts and suggestion for improvement.

Introduction

-Page 6 Line 5. The background sections of papers often simply state that physician empathy correlates with outcomes, but studies are increasingly demonstrating that physician characteristics (Tuerk et al., 2008, *Diabetes Care*), and more specifically physician empathy (Chaitoff et al., 2019, *Journal of General Internal Medicine*), may not be nearly as strongly associated with patient satisfaction or outcomes as once believed (at least not when empathy is self-reported, as is the case in many studies underpinning the empathy literature). Articles studying empathy need to avoid hyperbole in describing the importance of “physician empathy” (defined in any number of ways) in order to avoid becoming perspective pieces. I would encourage the authors to not simply cite entire books (written by members of a research group that has a clear financial conflict of interest with the JSE) or reviews to support their argument that empathy is important. Instead, they should make their argument in their introduction by including more quality primary literature while also noting the very real limits of our understanding of how empathy as it is currently measured impacts patient experience and clinical outcomes. Given the very real limitations of self-report empathy (namely that it is being increasingly shown not to correlate with meaningful outcomes), this is necessary to explain why this study of patient-report empathy is so important.

We completely with the reviewer and have made substantial changes in the introduction session. Although we kept the idea related to the importance of empathy in the first paragraph, in the following paragraphs we used more specific references and were clearer about the limitations of using self-assessed empathy and how fragile is the connection between self-assessed empathy and clinical outcomes.

The new introduction is below:

“INTRODUCTION

Physician empathy is crucial for patient care.^{1,2} Empathy enhances professional fulfillment,¹ diminishes physicians’ burnout,³ and is related to clinical competence.⁴ Empathy is vital to understand, acknowledge, and address patients’ needs in clinical encounters and to construct a shared and feasible therapeutic plan considering patients’ beliefs and context.^{5,6} Empathetic behaviour of health professionals is a cornerstone to establish meaningful conversations with patients, decrease anxiety and reveal patients’ emotions and beliefs relevant to patients’ experiences.⁷⁻⁹

The relevance of empathy to patient care has led to increasing calls to develop learning strategies to foster the capacity of physicians in training for empathic understanding.⁴ Unfortunately, research findings with undergraduate students have raised concerns over eventual negative impacts of medical schools on student empathy (for a discussion see Ferreira-Valente et al.¹⁰). Despite the controversy over the effect of medical school on empathy, it is consensual that an important limitation of empathy research has been the frequent reliance on participant self-assessments, in general using the Jefferson Scale of Physician Empathy (JSE)¹¹ or the Interpersonal Reactivity Index (IRI)¹². This is the case, for example, of longitudinal studies of medical student empathy¹³ or of studies on outcomes of interventions to develop empathy.¹⁴⁻¹⁶

Despite the international use of established questionnaires, it is unknown to what extent empathy self-assessment scores reflect empathic behaviours as observed by others, specially patients.¹⁷⁻¹⁹ The wider literature on self-assessments suggest these may be insufficient to capture the full breath of behaviors or attitudes.^{19,20} It is therefore of paramount importance to characterize associations between empathy assessed by oneself and by others.

Recently, a multi-centre study in Brazil¹⁹ of associations between self-reported and patient derived empathy measures for physicians, reported that those may be unrelated, suggesting that doctor empathy self-assessments were not indicative of empathy, as perceived by patients. It is thus likely that the same is the case for resident or student empathy, but this remains to be confirmed empirically. Moreover, a recent study showed a lack of correlation between self-assessed empathy by primary care physicians and clinical outcomes in patients with Diabetes²¹. The lack of correlation between self-assessed empathy and both patients' assessments and clinical outcomes are powerful arguments to expand the field towards including patients' perspectives.

In the current article, we sought out to investigate the correlation between medical students' and residents' self-assessed empathy levels with the empathy levels as perceived by the patients assisted by them directly in real clinical encounters. We also investigated whether patients' measures are valid and reliable tools to be used as assessment methods of the empathy levels of medical students and residents. We believe that understanding how patients perceive the empathy of medical students and residents in the context of real medical encounters can inform educational interventions to foster a more humanistic practice. Our research may help to enlighten the utilization of empathy measurements to guide educational practices.”

-Page 6 Lines 31-54 (paragraph 2). This paragraph is difficult to follow and seems to try to explain to the reader why therapeutic plans succeed or fail, which is beyond the scope of this paper. I would shorten this considerably or remove it entirely.

We agree with the reviewer and removed the paragraph.

-Page 7 Lines 29-36. Much of these sentences appears to be more appropriate for the discussion, I would remove from the introduction.

We agree with the reviewer and removed the sentences.

Methods

-I think a brief discussion of why the 4 scales were chosen would be helpful. Specifically, of the self-report empathy measurements, IRI was not developed to necessarily be healthcare specific as compared with the JSE, which was developed for use in healthcare populations. However, both the JSPPE and CARE patient-report empathy measures were developed with healthcare settings in mind. Why not select one of each type (a healthcare specific scale and one more general scale of empathy) for both self-report and other-report measures of empathy?

We agree with the reviewer on his concern about using different scales to measure the same construct. However, regarding the self-reported scales, there is literature from 5 countries showing that JSE and IRI scales measure different forms of empathy, so we decided to use all scales as we could not extrapolate interpretation of findings across instruments. The JSE is widely used to measure empathy in the medical context and IRI has a more broaden utilization.

Regarding the similarities between CARE and JSPPE, although they have the same objective, they were developed from different stances. The JSPPE was developed with the idea that empathy is a predominantly cognitive construct, while the CARE scale was developed using data collected from interviews with patients. JSPPE and CARE were related in our last study but, in this study, they were not correlated. We believe that those data are worth sharing to create awareness of the complexity of this field and limitation of those measures.

We made our choices explicit in the methods session by stating the following sentences:

“We used JSPPE because it shares the same concept of empathy as the JSE – both were developed by the same research group.²⁴ CARE was developed aiming the concept of empathy as

perceived by patients and there is evidence of the reliability, internal validity and consistency of this scale.²⁶”

“Previous studies had demonstrated that JSPPE and CARE were unidimensional with high reliability coefficients – respectively 0.88 and 0.97¹⁹ As the Portuguese version of these instruments had only been validated with a sample of medical doctors in Brazil, we have conducted new analysis to investigate whether the psychometric properties were similar for physicians in training.”

“We used the Physician and Student versions of the Jefferson Scale of Empathy (JSE),¹¹ and the Interpersonal Reactivity Index (IRI).¹² These are the two scales most extensively used in empathy research. JSE was developed specifically for healthcare contexts whereas IRI was developed for the general population.”

-The authors should add an analysis. Specifically, they should calculate correlation coefficients for the IRI vs JSE. To my knowledge, in Hojat et al.’s validation papers on the JSE, they never compare JSE to an accepted standard (like the IRI), and studies by other authors (eg Costa et al., 2017, Academic Medicine) that have done so have reported weak correlations between the two. If it is found that so much of the physician empathy literature may be based on a scale that doesn’t measure empathy as it is most commonly conceived by others in psychology/sociology, it would be very pertinent to report here.

Response: Thank you for your suggestion. We have calculated and added the correlation between IRI and JSE and their sub-scales.

WE ADDED: “We found a moderate and significant correlation between JSE and IRI ($r=0.44$, $p < 0.05$). We also found positive and significant correlations between the sub-scales ranging from weak to moderate magnitude (Table 2)

Table 2. Pearson correlations between JSE and IRI.

		IRI				
		Fantasy Scale	Perspective Taking	Empathic Concern	Personal Distress	IRI Total
J S E	Perspective Taking	0.355*	0.285*	0.632*	-0.048	0.485*
	Compassionate Care	0.364*	0.342*	0.603*	0.046	0.346*
	Standing in the Patient’s Shoes	0.318*	0.184	0.492*	-0.035	0.031
	Jefferson Total	0.033	0.038	0.240*	-0.183	0.435*

Abbreviations: JSE = the Jefferson Scale of Empathy (JSE); IRI = Interpersonal Reactivity Index.

We also added the discussion of those results in the discussion session:

“Contradictory to previous studies where weak correlations were found between JSE and IRI,²⁹ our results indicated a moderated correlation between the JSE and IRI, suggesting that both scales may be measuring the same aspect of empathy. Although this finding adds to the number of validity evidence of the JSE, the outcomes should be carefully interpreted since it seems that the relation may be related to the sample and context.”

-I wonder if the pearson correlation should be spearman correlations. From my experience, empathy scores are often not normally distributed; the authors might want to check a histogram of empathy scores to ensure appropriateness of the analysis.

Thank you for your comment. The absolute values of skewness and kurtosis for all sub-dimensions were within the acceptable range of the normal distribution (between -2 and 2) for the scales, with the exception of one sub-dimension Compassionate Care. Subsequent analyses demonstrated that there

was no difference when using the parametric or non-parametric analysis in the outcome. Therefore, we only present the outcomes of parametric analyses to facilitate the comparison between different studies and population.

-Page 10 Line 10-11. Physicians were informed of the nature of the research, which may bias the results (social desirability bias). This should be mentioned as a limitation in the discussion.

Response: Thank you for your comment. We have added a sentence in the limitation section about informing the participants.

We added the following sentence to the discussion: "A final limitation was that participants were informed of the nature of the research, which may have induced behaviors more socially desirable and have biased the results".

Results

-Would add two columns to table three to compare the JSE and IRI to each other (as described above).

Thank you very much for your suggestion. We have added a new table with the correlation of the JSE and IRI and their dimensions (as above).

-Would appreciate measures of spread be included throughout the results (SD or IQR, whichever is more appropriate).

Thank you very much for your suggestion. We have added the SD where appropriated in Tables 1 and 3.

Discussion

-I think the discussion about the inaccuracy of self-assessment should be expanded, perhaps discussing why the mismatch occurs.

We have expanded the discussion about the inaccuracy of self-assessment and trying to explain the reason of this mismatch.

We added the following paragraph to the discussion: "As expected, our results demonstrated that self-assessment of empathy by interns and residents does not correlate with patients' assessments, in alignment with the results previously observed for senior clinical practitioners.¹⁹ This finding corroborates the general literature that points out the inaccuracy of self-assessment.^{17 18} Overall, physicians in training might become overconfident over time. For example, consecutive participation in clinical practice may increase students' self-confidence^{32 33}, which does not necessarily predict their performance. Furthermore, self-assessment questionnaires in empathy often focus only on whether the participants are aware of how they have to behave to be empathic. However, knowing how to behave does not necessarily translates into a change of the behavior in practice. Our results are aligned with this possible mismatch between intention and action."

-Page 15, line 34. The authors state that the "CARE and JSPPE scales could be used as assessment tools and detect elements that may interfere in patients' perception of students and residents empathy." I would like more of a discussion of why this is an appropriate recommendation. This study shows that the JSPPE and CARE do not correlate with the IRI (a well-validated scale of empathy), and, to my knowledge, there has not been convincing evidence showing that either scale correlates with any meaningful clinical outcomes. As such, while the scales may have construct validity, I'm not sure what the purpose of using them in medical education assessment would be. If, however, the scales have been shown to correlate with a meaningful outcome (be they patient experience or clinical in nature), then that should be reported here in the discussion. This discussion might also

touch on the bigger issue with the empathy literature – lots of correlations with communication/patient experience outcomes more generally, but not many scales shown to correlate with clinical outcomes.

We have reformulated the initial paragraphs of the discussion to address the lack of correlation of self-assessment with clinical outcomes.

“In this study, we sought to investigate whether empathy self-assessment by physicians in training correlated with their patients’ assessments. We also evaluated the validity and reliability of the two instruments for patients’ assessments of physicians in training. Our findings corroborated the hypothesis that self-assessment of empathy by interns and residents did not correlate with patients’ assessments, in line with findings with senior clinical practitioners.¹⁹ Taking into consideration that this study used the 4 empathy scales– 2 self and 2 patient assessments – most widely used on empathy research, the findings are of particular relevance. A former study in five countries had demonstrated that the two self-reported scales did not capture the same empathy construct.²⁹ The lack of correlation between self- and patient assessed empathy has implications for how the literature on health professional’s empathy is interpreted. Quite likely, findings from studies using any of the 4 scales, which often compare empathy across studies, are not directly comparable. Like in the parable of the elephant and the blind man in which each blind man feels a different piece of the elephant, it is possible that such studies capture different elements of the complex psychological construct called empathy. To develop empathy studies relevant to inform medical education, it is crucial, at this moment, to clarify which scale – if any – offers measure which correlates with meaningful clinical or educational outcomes. For example, recently, Chaitoff et al.²¹ found that self-assessed empathy levels of primary-care physicians were not correlated with laboratorial outcomes in patients with Diabetes. This result enlightens the debate on the correlation between self-assessed empathy and clinical outcomes by showing that a relationship of cause-effect between those two variables is unlikely.

Empathy is a complex construct with cognitive, affective, behavioral and moral dimensions entailing different lenses to be fully understood.^{1 30} When patients are invited to discuss what is a positive outcome through their perspectives, the complexity increases. Patients’ definition of a positive outcome may vary along the course of their disease and life. We believe that empathy is a necessary psychological trait for the doctor to understand the singularity of each patient and individualize therapeutic plans in alignment with patients’ needs and beliefs. Considering both the complexity of empathy and the singularity of patients’ experiences, we invite researchers in this field to expand their focus.

First, longitudinal studies or pre/post-test evaluations of learning strategies using self-reported empathy cannot anticipate physicians in training performance on real clinical encounters. We agree that self-reported measurements can help teachers to start the conversation with students around the relevance of being empathic for becoming a caregiver. However, if educators want to mirror future performance, it is crucial to include real patients in the assessment of students. Specially during the transition to independent practice, when students face the challenges related to adapting to the constraints of the health care system, medical educators should reinforce the importance of empathy, while helping students to align theory and practice through effective role-modeling.³¹

Second, studies targeting the understanding of the possible association between empathy and clinical outcomes should take into consideration the importance of empathy on building a therapeutic alliance between the doctor and the patient. So, we hypothesize that punctual assessments by patients after a singular encounter are not enough to capture the phenomena under study and we should use instruments to measure the quality of the relationship between the doctor and patient. However, we do believe that punctual assessments may be efficient to give feedback to clinicians on their attitudes and behaviors nurturing the development of their communication skills.”

-The correlation between JSE and IRI should be reported in the results and a discussion of what is found should be reported in the discussion.

Thank you very much for your suggestion. We have included the correlation between JSE and IRI in the results section. We also included a discussion about this finding.

-The limitations can be expanded, including mentioning potential effects of discussing the purpose of the study with participants prior to providing them the scales.

We have expanded the limitation section including the potential effects of discussing the purpose of the study, lack of patients' perception, contextual environment and the having only one assessment of the physician in training.

The new limitation section follows: "A limitation of the present study is that we did not address the influence on patient perceptions of contextual or environmental factors, such as the consultation time, delay in the waiting room or the comfort of the environment. These elements may have influenced patients' perspectives.^{39 40} Another limitation was the study's inability to pair in time the assessments of physicians and patients. Also, the sample of physicians in training was not randomly selected. A final limitation was that participants were informed of the nature of the research, which may have induced behaviors more socially desirable and have biased the results."

VERSION 2 – REVIEW

REVIEWER	Alexander Chaitoff Cleveland Clinic Lerner College of Medicine of Case Western Reserve University
REVIEW RETURNED	26-Apr-2019

GENERAL COMMENTS	I appreciate the considerable work the authors put into this revision to address all the reviewer concerns. I think it is a very important piece of work, one of the more interesting studies in the medical empathy literature that I have read recently, and should help inform both researchers and educators.
---